# A Longitudinal Study of Breakthrough Cancer Pain: An Extension of IOPS-MS Study

**DOI:** 10.3390/jcm10112273

**Published:** 2021-05-24

**Authors:** Sebastiano Mercadante, Augusto Caraceni, Arturo Cuomo, Massimo Mammucari, Paolo Marchetti, Rocco Domenico Mediati, Silvia Natoli, Giuseppe Tonini

**Affiliations:** 1Anesthesia and Intensive Care & Pain Relief and Supportive Care, 00185 La Maddalena, Italy; 2Palliative Care, Pain Therapy and Rehabilitation, National Cancer Institute, IRCCS Foundation, 20133 Milan, Italy; augusto.caraceni@istitutotumori.mi.it; 3Anesthesiology, Resuscitation, and Pain Therapy Department, National Cancer Institute, IRCCS Foundation Pascale, 80131 Naples, Italy; a.cuomo@istitutotumori.na.it; 4Primary Care Unit, ASL RM1, 00193 Rome, Italy; massimo.mammucari@libero.it; 5Department of Clinical and Molecular Medicine, La Sapienza University of Rome, 00185 Rome, Italy; paolo.marchetti@uniroma1.it; 6Palliative Care and Pain Therapy Unit, Careggi Hospital, 50139 Florence, Italy; mediatir@aou-careggi.toscana.it; 7Department of Clinical Science and Translational Medicine—University of Rome Tor Vergata, 00133 Rome, Italy; silvia.natoli@uniroma2.it; 8Medical Oncology Department, Campus Bio-Medico University of Rome, 00128 Rome, Italy; g.tonini@unicampus.it

**Keywords:** cancer pain, breakthrough pain, opioids

## Abstract

The aim of this study was to longitudinally assess the characteristics of background pain and breakthrough pain (BTcP), analgesic treatment, and satisfaction with treatment four weeks after the first assessment. Methods: Adult cancer patients with a diagnosis of BTcP were included. At T0, age, gender, visit setting, cancer diagnosis, the extent of the disease, ongoing anticancer treatments, and Karnofsky level were recorded. The background pain intensity in the last 24 h (on a numerical scale 0–10), opioids used for background pain, and their doses, expressed as oral morphine equivalents (OME), as well as other analgesic drugs, were recorded. The number of BTcP episodes, their intensity, predictability and precipitating factors, onset duration of untreated episodes, and interference with daily activities were collected. Analgesics and doses used for BTcP, and the mean time to meaningful pain relief after taking medication, were assessed. The level of satisfaction with BTcP medication was also assessed. Adverse effects to be attributed to these medications were also recorded. At T4, the same data were evaluated. Results: After one-month follow-up, patients had a lower number of BTcP episodes and peak intensity, possibly due to the optimization of background analgesia. The principal characteristics of BTcP did not change significantly. Conclusion: A careful and continuous assessment should be guaranteed to all patients to limit the burden induced by BTcP, other than treating BTcP episodes with short-onset opioids.

## 1. Introduction

Pain is frequently reported by cancer patients, and the prevalence is reported to be variable according to the stage of disease, ranging from 20–30% during the initial stages and up to 75% in advanced disease. This symptom strongly affects the patients’ quality of life (HRQoL) and their daily activities [1]. Relieving pain in cancer patients is fundamental in the process of care and still remains a challenging priority for clinicians [2,3].

Cancer pain management can be particularly critical when patients face peaks of pain intensity, even though the background pain is adequately controlled by opioids given around the clock for most hours of the day. This phenomenon is commonly known as breakthrough cancer pain (BTcP) [4]. BTcP is a frequent condition involving about 60% of patients with cancer pain. BTcP is characterized by a typical temporal pattern, including a short onset (about 10 min) and duration (about 45 min/episode), with a moderate–severe intensity that is well distinguished by the background pain considered clinically acceptable and not requiring dose adjustment in opioid therapy. BTcP has an evident impact on the quality of life of patients [5].

In a study of BTcP with a sample of more than 4000 patients diagnosed with BTcP, a large percentage of patients (86%) experienced marked interference with their daily activities due to BTcP [6].

Indeed, BTcP is a chameleonic phenomenon that may change in individuals according to different clinical features, stages of disease, and treatments [7]. For example, in cancer patients followed at home, the prevalence of BTcP decreased after one month, possibly due to a progressive reduction in physical activity or as a consequence of better background pain control [8]. Most studies of BTcP are cross-sectional; thus, changes in time, also according to the possible clinical interventions, have rarely been assessed.

The aim of this study was to longitudinally assess the characteristics of background pain and BTcP, analgesic treatment, and satisfaction with treatment four weeks after the first assessment.

## 2. Materials and Methods

This paper is based on data from the IOPS-MS study, a prospective multicenter study conducted in 32 centers in Italy [6]. An extension study was planned from the original study. Patients who agreed or were able to be re-assessed subsequently (4 weeks after), were selected from the original sample.

### 2.1. Patients

Adult cancer patients were included in the IOPS-MS study [6]. Patients with both the initial evaluation (T0) and a follow-up visit after four weeks (T4) were taken into consideration. Patients were seen in different settings, as outpatients, inpatients, or in day hospital. Thirty-two centers participated in the study. Inclusion criteria were as follows: age ≥18 years; a diagnosis of cancer; stable and well-controlled background pain (pain intensity ≤4 on a 0–10 numerical scale); and the presence of BTcP episodes, clearly distinguished from background pain, with moderate–severe intensity. A pre-defined algorithm was used to make a diagnosis of BTcP [6]. Patients were excluded if they had a condition of unstable or uncontrolled background pain (>4/10), peaks of low pain intensity (<5/10), and were poorly collaborating.

### 2.2. Measurements

At T0, age, gender, visit setting, cancer diagnosis, the extent of the disease (loco-regional or metastatic), ongoing anticancer treatments, and Karnofsky level were recorded.

The background pain intensity in the last 24 h (on a numerical scale 0–10), opioids used for background pain, and their doses, expressed as oral morphine equivalents (OME), as well as other analgesic drugs, were recorded.

The number of BTcP episodes, their intensity (on a numerical scale 0–10), predictability and precipitating factors, onset (≤10 min or >10 min), duration of untreated episodes, and interference with daily activities (on a scale from 0 to 3: none, mild, much, and very much, respectively), were collected. Analgesics and doses used for BTcP, and the mean time to meaningful pain relief after taking medication, were assessed. The level of satisfaction (on a scale from 0 to 3: none, mild, much, and very much, respectively) with BTcP medication was also assessed. Adverse effects to be attributed to these medications were also recorded.

At T4, the same data were evaluated. During this interval, patients were visited or contacted by phone to eventually modify the analgesic regimen, according to their clinical needs.

### 2.3. Ethical Considerations

All patients provided written informed consent, and the committees for medical research ethics in each center approved the study before initiation.

### 2.4. Statistics

Patient, BTcP, background pain, and BTcP management characteristics at T0 and T4 were described using mean (and 95% confidence interval (CI)) or frequency (and percentage) as appropriate. For drug doses, the 95% CI was calculated only when at least five patients were receiving that drug. When necessary, 95% confidence intervals were censored at 0, and the upper limit was adjusted to ensure 95% coverage. Changes in BTcP, background pain, and BTcP management characteristics between T0 and T4 were assessed using McNemar’s chi-squared test or the paired two-sample t-test, as appropriate. The partially overlapping samples t-test [9] was used to compare the change between T0 and T4 doses of drugs that were administered at both time points to at least five patients (oxycodone SR, oxycodone + paracetamol, oxycodone + naloxone, fentanyl TD, buprenorphine TD, and OTFC). For other drugs that were administered at both time points to less than five patients (but were administered to at least five patients at one of the two time points), the unpaired two-sample t-test was used. The proportions of patients under each BTcP opioid were compared between the satisfied and unsatisfied patients using Fisher’s exact test. Analyses were performed using the statistical software R v4.0.2., and *p*-values <0.05 were considered statistically significant. The correlation of the change in background pain intensity between T0 and T4 with the change in intensity and number of BTcP episodes was evaluated using Spearman’s correlation coefficient.

## 3. Results

Of the original 4016 patients, 145 patients had a follow-up visit four weeks after the first evaluation. The characteristics of patients are described in Table 1. Fifty-eight patients (53%) were receiving anticancer therapy. Mucositis, candidiasis, and dry mouth were found in 8 (5.5%), 3 (2.1%), and 7 (4.8%) patients, respectively.

### 3.1. Background Pain

The pain mechanisms were mixed (*n* = 93, 64.1%), nociceptive (*n* = 43, 29.7%), and neuropathic (*n* = 9, 6.2%). The mean background pain intensity was 3.06 (CI 1.88–3.23) and 2.37 (CI 2.18–2.57) at T0 and T4, respectively. The difference was significant (<0.001). The mean daily OME was 98.3 mg (77.1- 119.4) and 102.8 mg (83.1-122.5) at T0 and T4, respectively. The difference was not significant (*p* = 0.73). Drugs and doses used for background pain are listed in Table 2.

### 3.2. Background Pain

The mean number of BTcP episodes/day was 2.5 (CI 2.3–2.7) and 2.1 (CI 1.9–2.3) at T0 and T4, respectively. The difference was significant (*p* < 0.001). BTcP intensity was 7.37 (CI 7.17–7.57) and 7.02 (CI 6.79–7.25) at T0 and T4, respectively. The difference was significant (*p* = 0.001).

There was a significant correlation between the change in background pain intensity (from T0 and T4) and the changes in BTcP intensity and the number of BTcP episodes (from T0 to T4) (r = 0.26, *p* = 0.002 and r = 0.29, *p* < 0.001, respectively).

At T0, BTcP was predictable and unpredictable in 89 (61.4%) and 56 (38.6%) patients, respectively. After four weeks (T4) data were stackable (*n* = 90, 62.1% and *n* = 55, 37.9%, respectively, *p* = 0.99).

BTcP onset was ≤10 min and >10 min in 130 (89.7%) and 15 patients (10.3%), respectively. The principal triggers for predictable BTcP were movement (*n* = 46, 31.7%) and swallowing (*n* = 9, 6.2%). Other triggers were cough (*n* = 7, 4.8%), procedures (*n* = 3, 2.1%), defecation (*n* = 2, 1.4%), and others (*n* = 3, 2.1%). Drugs and doses used for BTcP are listed in Table 3. At T4, the number of patients who were prescribed FPNS and FBT significantly increased in comparison with T0 (<0.001). 

The mean meaningful pain relief after BTcP medication was 12.8 (CI 10.4–15.3) and 12.7 (CI 10.3–15.1) minutes, at T0 and T4, respectively. The difference was not significant (*p* = 0.49). Adverse effects attributable to BTcP medications were minimal (see Table 4). At T4, the number of patients who were satisfied with BTcP medication increased, although these data did not attain any significance. A minority of patients changed the BTcP medication (Table 5).

## 4. Discussion

The main finding of this explorative study performed in patients who had a follow-up four weeks after the first assessment was that, after an initial assessment, the improvement of background pain was able to reduce the number of BTcP episodes and their intensity. Although the increase in opioid doses was not significant, comprehensive management of pain and BTcP was likely to benefit the patients. Thus, better basal analgesia may allow us to prevent the gravity of BTcP episodes. Optimization of opioid therapy has largely been considered fundamental in reducing the development of BTcP. Studies have shown that the higher the pain intensity, the higher the prevalence of BTcP [5,10], and that a refinement of opioid therapy may reduce the number and the intensity of pain peaks, particularly in patients with bone metastases and predictable BTcP on movement [11]. This is confirmed by a recent longitudinal analysis of patients with bone metastases, in which BTcP was related to a higher level of pain intensity. Moreover, patients with BTcP had a higher and worse pain intensity after one month [12]. Similarly, in a general cancer population, a significant association between the presence of BTcP and a higher average pain score after 2 weeks was reported [13].

In general, no substantial changes regarding the characteristics of BTcP were found, at least in a one-month period. This observation was reported in a concomitant study [14]. Of interest, in the latter study, in which BTcP treatment was carried out through one of the available rapid-onset transmucosal formulations, an improvement in some aspects of quality of life was longitudinally observed in a one-month period. This should be attributed to more continuous monitoring according to the time intervals dictated from the study and only partially to better efficacy of BTcP medication [15]. Moreover, about half of patients experienced a large number of BTcP at initial assessment, which decreased after careful clinical management, confirming that many patients required better background pain control. Thus, taken together, these data confirm that accurate care may globally increase patients’ satisfaction.

The second point is that the type of BTcP medications did not notably change, unless for FPNS and FBT. Opioid doses were slightly increased, as it occurred with opioids used for background pain. Of interest, the level of satisfaction with BTcP medication did not change and was considered acceptable in even more patients after one month. This observation confirms that BTcP medications maintain their effectiveness in time. Of interest, the initiation of fentanyl products as BTcP medication was associated with a reduced time for meaningful pain relief just after a week and maintained for one month, although doses of opioids for both background pain and BTcP were unavailable [15]. Few longitudinal studies of BTcP have been published. In an open-label continuation phase of BTcP patients, functioning, mood, and overall satisfaction were improved by a transmucosal fentanyl preparation for the management of BTcP for the long term [14]. In another longitudinal multicenter study performed in a very advanced cancer population with and without BTcP, an increase of 45% in OME was found, with a significant difference between patients with and without BTcP. Moreover, a decrease in the frequency of BTcP and the mean number of attacks was observed from baseline to day 28 [16]. In advanced cancer patients followed at home, the prevalence of BTcP decreased after one month, possibly due to a progressive reduction in physical activity or as a consequence of better background pain control [8]. Indeed, in another multicenter study of patients recruited in an oncological setting, the decrease of background pain intensity was associated with a significant decrease in the prevalence of BTcP over time [17].

### Strength and Limitation

Longitudinal analyses with repeated measures, as performed in the present study, increase the analytical strength of observations because of the individual changes in background pain, BTcP, and opioid drugs used. The sample size for a longitudinal study of palliative cancer patients is relatively acceptable. Thus, the results from this study allow clinicians to have more information to treat patients suffering from BTcP.

The study has some limitations. Only some of the patients recruited in the original study approved or were able to have a follow-up visit four weeks after the first evaluation. Thus, data should be considered explorative. This inevitable problem, often encountered in palliative care patients, could have biased the results. Second, patients were included in the study at different time points in their disease trajectory. For example, patients could be still receiving anticancer therapy in an oncologic setting or were receiving only palliative supportive care, being considered off-therapy. On the other hand, this reflects the real world in which most patients are followed. There is no standardized starting point for pain development, and this remains a challenge in cancer pain studies [12]. Third, this study did not provide data regarding the prevalence of the BTcP phenomenon, as patients were already recruited with a diagnosis of BTcP, according to a specific algorithm, shared in an investigator meeting performed before planning the original study [7].

The strong point of this study is that information was gathered from the real world, as this reflects the spontaneous clinical activity of experienced people working with cancer pain in different settings.

## 5. Conclusions

After an initial assessment, cancer patients with a diagnosis of BTcP had a lower number of BTcP episodes and peak intensity, possibly due to better comprehensive management, while the principal characteristics of BTcP did not seem to change significantly, at least in a one-month period. This suggests that careful and continuous assessment should be guaranteed to all patients to limit the burden induced by BTcP. There is a need for a longitudinal assessment of a phenomenon that is invariably dependent on the stage of disease, patient characteristics, and therapeutic interventions. Further longitudinal studies on a large scale should be performed to confirm this preliminary information.

## Figures and Tables

**Table 1 jcm-10-02273-t001:** Characteristics of patients (*n* = 145).

Age (years), mean (95% CI)	62.5 (60.5, 64.6)
Gender (M/F)	85 (58.6%)/60 (41.4%)
Karnofsky, mean (95% CI)	68 (65.71)
Primary tumor	
lung	47 (32.4%)
breast	15 (10.3%)
prostate	14 (9.7%)
colon–rectum	12 (8.3%)
head–neck	7 (4.9%)
pancreas	6 (4.1%)
liver	6 (4.1%)
stomach	5 (3.4%)
gynecological	5 (3.4%)
others	36 (24.1%)

**Table 2 jcm-10-02273-t002:** Number of patients receiving opioids and mean doses (mg/die) (95% CI) used for background pain.

Analgesic Treatments	T0	T4	*p* Value
**Non-opioid analgesics** (paracetamol or anti-inflammatory drugs)	47 (32.5%)	46 (31.8%)	1.00
**Opioids**			
Hydromorphone	4 (2.8%)	5 (3.4%)	1.00
mg/die, mean (95% CI)	7 (NA)	22 (0–47)	0.19
Oral morphine SR	2 (1.4%)	6 (4.1%)	0.13
mg/die, mean (95% CI)	473 (NA)	230 (0–582)	0.67
Oxycodone SR	29 (20.0%)	29 (20.0%)	1.00
mg/die, mean (95% CI)	41 (23–58)	51 (29–73)	0.41
Codeine + paracetamol	5 (3.4%)	3 (2.1%)	0.62
mg/die, mean (95% CI)	84 (53–115)	80 (NA)	0.87
Oxycodone + paracetamol	14 (9.7%)	12 (8.3%)	0.48
mg/die, mean (95% CI)	37 (26–48)	44 (32–56)	0.28
Oxycodone + naloxone	43 (29.7%)	41 (28.3%)	0.79
mg/die, mean (95% CI)	42 (33–52)	42(34–49)	0.94
Tapentadol	5 (3.4%)	6 (4.1%)	1.00
mg/die, mean (95% CI)	210 (2–418)	208 (51–365)	0.99
Tramadol	2 (1.4%)	2 (1.4%)	1.00
mg/die, mean (95% CI)	295 (NA)	750 (NA)	NA
Oral morphine (IR)	4 (2.8%)	5 (3.4%)	1.00
mg/die, mean (95% CI)	46 (NA)	60 (0–130)	0.73
IV morphine	5 (3.4%)	3 (2.1%)	0.51
mg/die, mean (95% CI)	110 (10–210)	133 (NA)	0.76
SC morphine	1 (0.7%)	3 (2.1%)	0.62
mg/die, mean (95% CI)	10 (NA)	20 (NA)	NA
Methadone	1 (0.7%)	1 (0.7%)	1.00
mg/die, mean (95% CI)	6 (NA)	6 (NA)	NA
Fentanyl TD	35 (24.1%)	36 (24.8%)	1.00
µg/h, mean (95% CI)	59.3 (48.8–69.8)	63.8 (52.2–75.3)	0.49
Buprenorphine TD	8 (5.5%)	5 (3.4%)	0.37
µg/h, mean (95% CI)	48.1 (35.2–61.1)	58.5 (28.2–88.8)	0.44

NA = not applicable the calculation of 95% CI.

**Table 3 jcm-10-02273-t003:** Opioids and doses used for BTcP at T0 and T4. OTFC = oral transmucosal fentanyl citrate, FBT = fentanyl buccal tablet, FBST = sublingual fentanyl, FPNS = fentanyl pectin nasal spray, INFS = intranasal fentanyl, IR = immediate release morphine, SC Morphine = subcutaneous morphine, IV = intravenous morphine, NA = not applicable the calculation of 95% CI.

	T0	T4	*p* Value
OTFC	6 (4.1%)	6 (4.1%)	1.00
µg, mean (95% CI)	350 (55–645)	420 (166–674)	0.64
FBT	35 (24.1%)	38 (26.2%)	0.55
µg, mean (95% CI)	234 (181–288)	272 (222–322)	0.04
FBST	15 (10.3%)	18 (12.4%)	0.55
µg mean (95% CI)	200 (102–298)	206 (118–293)	0.88
FPNS	26 (17.9%)	46 (31.7%)	<0.001
µg, mean (95% CI)	154 (114–164)	191 (154–228)	0.06
INFS	1 (0.7%)	1 (0.7%)	1.00
µg, mean (95% CI)	100 (NA)	400 (NA)	NA
IR morphine	25 (17.2%)	18 (12.4%)	0.07
mg, mean (95% CI)	10 (8–12)	10 (8–13)	0.91
SC morphine	1 (0.7%)	4 (2.8%)	0.37
mg, mean (95% CI)	5 (NA)	11 (NA)	NA
IV morphine	3 (2.1%)	3 (2.1%)	1.00
mg, mean (95% CI)	13 (NA)	18 (NA)	NA
Others	19 (13.1%)	13 (10.3%)	0.48

**Table 4 jcm-10-02273-t004:** Adverse effects attributable to BTcP medications.

Adverse Effects	T0	T4	*p* Value
Headache	0 (0.0%)	0 (0.0%)	1.00
Confusion	1 (0.7%)	0 (0.0%)	1.00
Gastralgia	0 (0.0%)	1 (0.7%)	1.00
Nausea	0 (0.0%)	0 (0.0%)	1.00
Pruritus	0 (0.0%)	1 (0.7%)	1.00
Vomiting	0 (0.0%)	0 (0.0%)	1.00
Other	0 (0.0%)	0 (0.0%)	1.00

**Table 5 jcm-10-02273-t005:** Satisfaction with BTcP medication.

Satisfaction	T0	T4	*p* Value
Unsatisfied–indifferent	27 (21.4%)	5 (13.7%)	0.73
Satisfied–much satisfied	99 (78.6%)	120 (86.3%)	
Missing	19	6	
**Changes in BTP medication**			
No		132 (91.0%)	
Yes		13 (9.0%)	

## Data Availability

Data are available on request.

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
