# Peer review of "A Longitudinal Study of Breakthrough Cancer Pain: An Extension of IOPS-MS Study"

_jcm, 2021, doi:10.3390/jcm10112273_

Round 1

Reviewer 1 Report

The aim of this study was to longitudinally assess the characteristics of background pain and breakthrough pain (BTcP), analgesic treatment (mostly opiates), and satisfaction with treatment four weeks after the first assessment. To perform this assessment, of the original 4016 patient with BTcP at T0 (week 0) 145 patients had a follow up visit at T4 which was four weeks after the first evaluation.

From the present study the authors conclude that after an initial assessment cancer patient with a diagnosis of BTcP, patients have less number of BTcP episodes and peak intensity, possibly due to a better comprehensive management, while the principal characteristics of BTcP seem to do not change significantly, at least in one month period. The authors suggest that a careful and continuous assessment should be guaranteed to all patients to limit the burden induced by BTcP. There is a need for a longitudinal assessment of a phenomenon that is invariably dependent on the stage of disease, patient characteristics, and therapeutic interventions. Further longitudinal studies on a large scale should be performed to confirm this preliminary information.

A major issue with this study is that although the authors started with 4016 patients with BTcP at time 0 (T0) only 145 patients of the original 4016 patients were assessed at the 4 week period (T4). To further complicate matters, the 145 remaining patients examined at T4 are a very heterogenous group of cancer patients (10+ different cancers). Given that relatively few patients made it from T0 to T4 and the heterogeneity of types and stages of cancer included in this study, is very difficult for a reader understand what results achieve significance and whether a result is significant or not is due to; the type of cancer, the low “n” in many of the cancers assessed at T4, whether different mechanisms drive BTcP in different cancers, evolution of disease, etc.

While few in the cancer pain community would argue with these conclusions, few of these conclusions are based new, statistically significant data that is presented in this manuscript. Thus, it is not clear how this manuscript moves the field of cancer pain forward in terms of; understanding the mechanism that drive BTcP, how BTcP evolves with the underlying cancer, and how to best manage BTcP in patients with different types of cancer. More clearly addressing the above issues would significantly increase the impact of this manuscript.

Author Response

1.The reviewer is right. However, the study remains one with the largest number of patients assessed longitudinally, rather than with an usual one point. Although cancer diagnosis, obviously cannot discrimitate patients for the relatively low number of patients, as it occurred in cross-sectional design of the original study, the study provided other informations regarding the clinical changes, which weres the primary objective of the study: "The aim of this study was to longitudinally assess the characteristics of background pain and breakthrough pain (BTcP), analgesic treatment, and satisfaction with treatment four weeks after the first assessment 

2.This study did not have these objectives, given the design of study, which was observational. The treatments were discretional (see protocol of the original study). Please, consider that it is not too esay to have a follow-up of patients with (mostly) advanced cancer.

Reviewer 2 Report

The study reports the results of a 4-weeks follow up period of patients with breakthrough cancer pain (BTcP).

Several parts of the article are unclear and should be corrected first.

1. My major concern is that it is to me unclear what the difference is between T0 and T4, except for the fact that they are 4 weeks apart.  If any intervention happend at T0, or patients were just experiencing BTcP, a four weeks follow up has sense.  The authors should be more clear about this. For example, if the period that the patients are experiencing BTcP or the tumor stage at T0 varies from patient to patient, the follow up of 4 weeks would examine the patients at different stages of the tumor or BTcP stage and influence the results.  

2. Statistical analysis (1):  the authors used t-tests.  These tests require that the data are normally distributed around their mean.  The authors should check for normality and mention this in the article. 
3. Statistical analysis (2):  a lot of P-values were calculated. Was any correction for simultaneous hypothesis testing applied ?

4. Results:  it is unclear to me what the totals are and how the different percentages have been calculated.
The first paragraph of the results section mentions 145 patients.
Table 1 shows, for the primary tumor, 153 patients, while the percentages have been calculated with 145 as total.  
Table 2 shows a certain number of patients for the different analgesics.  The number of patients included does not sum op to 145, while the percentages are.
Table 3 : similar remark as for Table 2.
Table 5:  The total number of patients for T4 is 131, not 145.

5. Results , section 3.1:  the mean background paint intenisty was 3.06 (CI 1.88 - 3.23).  The value of 3.06 is not centered in the confidence interval, while the applied statistical tests result in symmetrical confidence intervals.  Please correct or clarify.

6. Table 2:  please write mg/day instead of mg/die.

7. Table 2:  I understand that confidence intervals were not given when ther were less than 5 patients involved.  However, what does the 0,47 mean for hydromorphone at T4, or the 0.582 for oral morphine sr at T4 ? 

8. Table 2:  the confidence interval for oral morphine (ir) at T4 (0-130) is not centered around the mean value of 60.  Please clarify or correct.

9. Discussion, section 4.1, end:  The authors write 'Authors should discuss the results and how they can...  also be highlighted'.  I presume that this is a part that comes from a text describing how a discussion should be written, and is not part of the article.

Introduction:  ... patients (86%) experenced:  experenced should be experienced
Results:  section 3.1:  The was a significant correllation:  The should be There

Author Response

  1. The differences were reported in tables 2 and 3 and along the text  Of course something (for example predictability), did not change, while intensity and frequency of BTcP did. This was exactly the aim and the sense of the study. Clealry at T0 treatment profile (drugs and dosing) changed according to local policy and the contiunous assessment through the weeks before the final assessemnet at T4, as described in Results, either  for background pain and BTcP . The changes of the stage was not taken into consuideration, considering the short time intereval for a cancer trajectory of years, also because most of them were "advanced".  Longer studies, really difficult to perform in this setting, could provide more insights (see conclusion)..   
  2. The majority of variables for which a t-test has been used appeared to be normally distributed (according to the Shapiro-Wilk test for normality). For consistency we decided to use t-test for all comparisons involving continuous variables
    3.  we agree with the reviewer that correction for multiple testing would have been preferable. However, given the sample size of our study correction for multiple testing would result in an inadequate statistical power. Since this and that our analysis is explorative in nature, we did not use adjustment for multiple comparisons.

4. Results:  it is unclear to me what the totals are and how the different percentages have been calculated.
The first paragraph of the results section mentions 145 patients. In tables mulitple answers were possible, so that the net sum may not correspond

5. Results ,

Done.

6. Table 2:  please write mg/day instead of mg/die.

Done

7. Table 2:  I understand that confidence intervals were not given when ther were less than 5 patients involved.  However, what does the 0,47 mean for hydromorphone at T4, or the 0.582 for oral morphine sr at T4 ? 

8. Table 2:  the confidence interval for oral morphine (ir) at T4 (0-130) is not centered around the mean value of 60.  Please clarify or correct.

7. Table 2:  I understand that confidence intervals were not given when ther were less than 5 patients involved.  However, what does the 0,47 mean for hydromorphone at T4, or the 0.582 for oral morphine sr at T4 ?

 (0, 47) and (0, 582) are the 95% confidence intervals for the average dose (in mg/die) of hydromorphone and oral morphine sr, respectively, at T4. Please note that less than 5 patients were receiving these drugs at T0 and, therefore, the 95% confidence interval is not provided for average doses at T0. However, the number of patients receiving these two drugs at T4 was 5 and 6, respectively, so 95% confidence intervals were provided for average doses at T4.

Moreover, the small number of data points for some drugs and the high variability across patients, could have been resulted in the lower limit of the 95% confidence interval being negative. This happened for some 95% confidence intervals (including those for hydromorphone and oral morphine sr average doses at T4). For these, we replaced the lower limit with 0, since negative doses are not possible.

 8. Table 2:  the confidence interval for oral morphine (ir) at T4 (0-130) is not centered around the mean value of 60.  Please clarify or correct.

See above (point 7).

9. Discussion, section 4.1, end:  The authors write 'Authors should discuss the results and how they can...  also be highlighted'.  I presume that this is a part that comes from a text describing how a discussion should be written, and is not part of the article.

deleted, thanks.

Introduction:  ... patients (86%) experenced:  experenced should be experienced
Results:  section 3.1:  The was a significant correllation:  The should be There

Done

Round 2

Reviewer 2 Report

In despite of the author's reply and some changes to the manuscript, I still have a few questions or remarks. Points 1, 3, 7, 8 should be addressed further, points 2, 4, 5, 6, 9 and the type are resoved. 
1.  It is still unclear to me what the difference is between T0 and T4.  Let me try to explain my confusion by one simple question: Did the patients start taking the described medication at T0, or did they take this medication already before ?
2. Statistical test: I agree
3. Correction for simultaneous hypothesis testign: I did not find in the manuscript that the study is explorative.  This should be mentioned in the discussion.
4. Totals and percentages I agree
5. I agree
6. I agree
7. In table 2, the separator used between the limits of the confidence interval are sometimes a comma (0,47 for hydromorphone), sometimes a dit (0. 582 for oral morphine sr) and sometimes a minus sign (29-73 for oxycodone sr and others).  All confidence intervals should be written with the same separator.
8. I am surprised to read that, on the one hand, data are normally distributed, and on the other hand, 95% confidence intervals have a lower limit below 0.
I have two questions/remarks:
8a. Are these confidence intervals the confidence intervals from the population, or the confidence intervals aroudn the mean ?
8b. Censoring confidence intervals at 0 is allowed, as long as the upper limit is changed to, in order to have a coverage of 95%. For example, if there is a probability of 2% between the calculated lower limit of the confidence interval and 0 and the upper limit remains unchanged, the confidence interval becomes a (95-2=)93% confidence interval.  The upper limit should be recalculated in order to have a 95% confidence interval.  I do not find any information about this recalculation in the manuscript.  The authors should mention this if they performed the recalculation, or, in case they did not do it, perform the recalculation and mention it is the materials and methods.
9. I agree

typo: I agree

Author Response

  1.  It is still unclear to me what the difference is between T0 and T4.  Let me try to explain my confusion by one simple question: Did the patients start taking the described medication at T0, or did they take this medication already before ?

See in methods: Patients with both the initial evaluation (T0) and follow-up visit after four weeks (T4). Please, consider that at T0 drugs were those previsouly used at time of the first evaluation, and at T4 drugs were those possibly changed during the intercourse T0-T4, according to the clinical needs, either for background pain and BTcP
3. Correction for simultaneous hypothesis testign: I did not find in the manuscript that the study is explorative.  This should be mentioned in the discussion.

I added this point at the start of discussion

  1. In table 2, the separator used between the limits of the confidence interval are sometimes a comma (0,47 for hydromorphone), sometimes a dit (0. 582 for oral morphine sr) and sometimes a minus sign (29-73 for oxycodone sr and others).  All confidence intervals should be written with the same separator.

Done

  1. I am surprised to read that, on the one hand, data are normally distributed, and on the other hand, 95% confidence intervals have a lower limit below 0.
    I have two questions/remarks:
    8a. Are these confidence intervals the confidence intervals from the population, or the confidence intervals aroudn the mean ?

Those are the 95% conficende intervals for the mean, censored when the lower limit of the interval is below 0. As the sample size for some of these confidence intervals is relatively low (i.e. less than 10 patients), standard errors are large resulting in wide confidence intervals that could stretch out to include negative values, if not censored.

Please, look at: When necessary, 95% confidence intervals were censored at 0 and the upper limit was adjusted to ensure 95% coverage.” 

8b. Censoring confidence intervals at 0 is allowed, as long as the upper limit is changed to, in order to have a coverage of 95%. For example, if there is a probability of 2% between the calculated lower limit of the confidence interval and 0 and the upper limit remains unchanged, the confidence interval becomes a (95-2=)93% confidence interval.  The upper limit should be recalculated in order to have a 95% confidence interval.  I do not find any information about this recalculation in the manuscript.  The authors should mention this if they performed the recalculation, or, in case they did not do it, perform the recalculation and mention it is the materials and methods.
The reviewer is right as the upper limit of those confidence intervals have been adjusted to ensure a 95% coverage. We now mention this in the statistical section of the paper.